# Differences in the Perception of Snacks and Beverages Portion Sizes Depending on Body Mass Index

**DOI:** 10.3390/nu17132123

**Published:** 2025-06-26

**Authors:** Anna Celina Durma, Maja Sosnowska, Adam Daniel Durma, Adam Śmiałowski, Leszek Czupryniak

**Affiliations:** 1Department of Diabetology and Internal Medicine, Medical University of Warsaw, 02-097 Warsaw, Poland; 2Department of Endocrinology and Radioisotope Therapy, Military Institute of Medicine—National Research Institute, 04-141 Warsaw, Poland; 3Faculty of Medicine, Warsaw University, 09-400 Warsaw, Poland; 4Department of Endocrinology and Metabolic Diseases, Polish Mother’s Memorial Hospital, Research Institute, Medical University of Lodz, 93-338 Lodz, Poland

**Keywords:** obesity, snacks, portion size, eating habits, obesity cause, estimation of calories

## Abstract

Introduction: The major cause of obesity is excessive calorie intake. Inappropriate dietary habits, like increased meal frequency, portion sizes, or amount of snacks consumed contribute to obesity development. Potential differences in the perception of snacks by people with different BMIs may be one of the causes of obesity. Assessment of the perception of snacks by people with excessive and normal body weight will allow us to check whether this parameter actually has an impact on the development of obesity. Materials and Methods: A survey study was conducted to check differences in assessing different snacks and beverages by individuals with varied BMIs. Participants analyzed snacks on presented photographs and assessed portion sizes, estimated caloric content, assessed whether the meal was healthy, and determined whether the indicated portion would be sufficient to satisfy their hunger. The study population was divided according to body mass index (BMI) into individuals with normal weight, overweight, and obesity. Additionally, the study group was divided according to gender and age. Results: There were no statistically significant differences in the majority of the studied parameters concerning BMI; however, the study revealed relatively low education level regarding caloric assessment. Conclusions: BMI seems not to have an influence on calorie and portion size perception of snacks. The majority of the population wrongly assessed the calorific value of snacks, which might contribute to obesity development. People have a tendency to overestimate the caloric value of snacks. Women assess the portion size of highly processed snacks as larger than men do.

## 1. Introduction

Obesity is one of the most significant health issues in modern and developing countries. The primary cause of obesity is an excessive energy intake—one exceeding energy expenditure. The human body can tolerate a positive energy balance, making it challenging to maintain a healthy weight, especially in an environment promoting calorie consumption. This phenomenon may be partially explained by evolutionary processes that have historically favored the consumption of high-fat and high-carbohydrate meals due to their high energy density, facilitating energy storage during periods of food deficiency [1]. Another factor contributing to a positive energy balance is unhealthy dietary patterns. Examples of such behaviors include frequent snacking on highly processed foods, consuming sugary beverages, eating while watching television or using a computer, and frequent dining out [2]. Snacks are often rich in saturated fats and simple carbohydrates, enhancing their palatability, thus increasing energy intake per serving [3,4]. Additionally, consuming snacks is associated with a sense of reward, which reinforces this behavior [5,6,7]. Usually snacking constitutes a significant proportion of total energy intake, partly due to the high availability of snack foods. A study by Myhre et al. found that snacks are an important part of the diet, accounting for 17–21% of energy intake [8]. A study on snack consumption in Finland showed that the share of energy from snack consumption was as much as 40% [9]. Snacks are often easier to purchase and consume than full meals [10]. Common snack items include sweets, salty snacks, desserts, sugary beverages, and alcoholic drinks [8,11]. Nevertheless, it is important to note that not all snacks are unhealthy. Snacks such as vegetables and some fruits can provide fiber, vitamins, and micronutrients that are essential for maintaining a healthy diet [12,13]. However, these healthier options tend to be less popular, particularly among young people [14,15], possibly due to lower palatability compared to processed foods. Fruits and vegetables are generally less sweet than processed snacks, which may contribute to their lower preference among consumers [14,16]. It may be an evolutionary tendency to choose sweet foods, which usually tend to provide more energy [14]. The tendency to consume more sweets than fruits and vegetables is particularly evident among children [17]. In recent years, there seems to be a change in this aspect. A study published in 2024 described an increase in the consumption of vegetables and fruits in 10 of 36 European countries and a significant decrease in the consumption of sweets in 12 of 36 European countries among adolescents [18]. The assessment of eating habits among children and adolescents and their possible change is very important, because habits which are determined in the early years of life often persist in adulthood [19]. Another factor contributing to excessive energy intake is portion size [20]. Studies indicate that in recent years, the portion sizes of served meals have significantly increased, including snack portions. This trend leads to higher overall food intake, which contributes to obesity [21]. Conversely, reducing snack portion sizes has been shown to decrease consumption [22]. On the other hand, it seems that smaller snack packaging may encourage more frequent purchases.

Research suggests that consumers often underestimate their actual portion sizes, highlighting the importance of portion awareness in obesity management [23]. The ability to accurately estimate caloric intake is crucial for maintaining a healthy body weight. A low-calorie diet remains one of the fundamental strategies for combating obesity [24]. Additionally, awareness of individual daily caloric needs plays a critical role in dietary control and establishing an effective weight reduction plan [25]

Clinical experience suggests that excessive snack consumption is a contributor to overweight and obesity; thus, our study aimed to evaluate whether the perception of snack size, satiety levels after consumption, and estimated caloric content vary based on an individual’s body mass index (BMI) and whether significant differences exist among selected subgroups. Finding a relationship between obesity and these variables could lead to a better understanding of the causes of the disease and more effective treatment.

## 2. Materials and Methods

An anonymous questionnaire was conducted among 205 volunteers aged 18 to 77 years. The questionnaire was made available on several websites dedicated to healthy nutrition and weight management or on a social media, and all adult visitors to these sites were invited to participate. We have also encouraged patients in outpatient clinics to fill out the forms. Volunteers were recruited by the Internet or in outpatient clinics (like general practice clinics, dietetic clinics, diabetology clinics, and endocrinology clinics). Participation in the study was open to all adult members of these communities, and no exclusion criteria were applied based on weight-related disorders (e.g., diabetes, hypertension), level of education, socioeconomic status, or age (provided participants were over 18 years old). We deliberately chose not to implement strict inclusion or exclusion criteria, as the study was conducted anonymously, and all health-related data were self-reported, making verification impossible. The questionnaire was conducted online, allowing participants an unlimited amount of time to respond to all questions.

A detailed characterization of the study population is presented in Table 1.

The questionnaire consisted of two sections. The first section included questions regarding basic participants’ data and eating behaviors, while the second, more extensive section comprised 15 color photographs of snacks and beverages. These meals were specifically prepared and photographed for this study by one of the authors. The authors chose international snacks and beverages, which are known and easily available in the most countries in the world. Efforts were made to diversify the snacks assessed. The survey included approximately equal amounts of highly processed snacks, minimally processed snacks (so-called “healthy snacks”), and various liquid snacks—beverages. Each photograph was accompanied by a detailed description of the dish, specifying ingredients that might not be visually apparent. To facilitate portion size estimation, standard-sized cutlery was placed next to the uniformly sized plates in all images. In addition, each photograph had the same, black background (Figure 1).

Participants were asked to subjectively assess portion size using an ordinal scale from 1 to 10, where “1” represented a very small meal and “10” indicated a very large meal. Additionally, they evaluated whether they would feel full after consuming the portion (yes/no) and estimated the time until they would feel hungry again (in hours). Satiety was further assessed on a 1–10 scale, where “1” indicated that the participant would still feel hungry after eating, and “10” signified maximum satiety. Participants estimated the time until they would experience hunger again by selecting one of the following options: less than 1 h, 1–2 h, 2–3 h, or more than 3 h.

Finally, respondents were asked to estimate the caloric content of each meal by providing a numerical value representing their best approximation of the absolute caloric content. In the results we also assessed the percentage of participants who accurately, which means 90–110% exact caloric value, estimated caloric content of presented snacks.

According to the authors’ assessment, the minimum time required to complete the questionnaire was about 35 min.

### 2.1. Evaluated Meals and Snacks

Among the assessed meals, 15 items were categorized as snacks:Highly processed snacks:Croissant with black coffee (no sugar);Cheesecake;Milk chocolate;Shortbread cookies with cream filling;Donut with rose jam filling;Hot dog with mustard and pork sausage.Minimally processed snacks:A handful of raisins;A handful of walnuts;Baby carrots;Apple.Beverages:A shot of vodka;Red wine;A bottle of beer;A glass of Coca-Cola;A glass of “sugar-free” orange juice.

The questionnaire, along with photographs of the meals, is included as Appendix A of this study.

In our study, we consider a snack to be a meal that is relatively small, does not require much time to consume and prepare, is easily available, and is relatively affordable. Our clinical experience shows that patients often overlook sweet drinks as a source of energy in their diet. Seeing this problem, we decided to include a section on beverages in the survey.

### 2.2. Statistical Analysis

Statistical analysis was performed using SPSS Statistics (IBM v23). To verify the assumption of normal distribution, the Shapiro–Wilk test was conducted. Quantitative variables were presented as mean (M) and standard deviation (SD) for normally distributed data, or as median (Med.) and interquartile range (IQR) for non-normally distributed data and ordinal variables. Categorical variables were analyzed using the Fisher exact test. Differences between study groups were assessed using the Kruskal–Wallis test. The odds ratio (OR) for meal overestimation was calculated by dividing participants into two groups: those with excessive body weight (BMI ≥ 25 kg/m^2^) and those with normal body weight (BMI < 25 kg/m^2^). Evaluation of the effect of age on the assessment of portion size and the caloric value of snacks was calculated using unpaired *t*-tests. The average age of the study group was approximately 40 years; thus, in age analysis subjects were divided into two subgroups depending on age: <40 years and ≥40 years. In cases of statistically significant results, appropriate post hoc tests were performed. A *p*-value < 0.05 was considered statistically significant.

## 3. Results

### 3.1. Description of Study Population

The questionnaire was conducted among 205 volunteers aged 18 to 77 years. The majority of participants were females (73%) with 27% being male responders. The body weight of participants ranged from 44 kg to 170 kg. The mean age of participants was 39.7 years. The mean BMI was 27 kg/m^2^. The studied group was divided into three groups: BMI < 25 where 92 participants belonged, 66 participants belonged to the group with BMI 25–29.9, and 47 participants belonged to the group with obesity. About 70% of volunteers were well educated and declared academic education. Details of the study group are presented in Table 1.

### 3.2. Estimation of Caloric Value

The results of the statistical analyses are presented in Table 2, Table 3, Table 4 and Table 5. No statistical significance was found for any of the analyzed parameters in the case of a croissant with black sugarless coffee, cheesecake, a glass of wine, cream-filled cookies, a hot dog, baby carrots, a jam-filled donut, a glass of Coca-Cola, an apple, or a glass of orange juice.

A statistically significant difference was observed in the estimated caloric value of a shot of vodka (*p* = 0.011). Post hoc analysis revealed a significant difference between individuals with a BMI < 25 kg/m^2^ and those classified as obese (*p* = 0.003), as well as between overweight and obese individuals (*p* = 0.044).

Similarly, statistical significance was found in the estimated caloric value of half a chocolate bar. The median estimated caloric value was 300 kcal for both individuals with a BMI < 25 kg/m^2^ and those classified as obese, while the median value for overweight individuals was 250 kcal, resulting in a difference of approximately 50 kcal. Post hoc tests indicated statistical significance between groups with a BMI < 25 kg/m^2^ and obesity (*p* = 0.034), as well as between overweight and obese individuals (*p* = 0.003).

A statistically significant difference was also observed in the estimated caloric value of a handful of nuts. The median estimated caloric value was 250 kcal for individuals with a BMI < 25 kg/m^2^, 210 kcal for overweight individuals, and 350 kcal for obese individuals. Post hoc analyses revealed statistically significant differences between groups with normal weight and obesity (*p* = 0.011), as well as between overweight and obese individuals (*p* = 0.013). The difference in median estimated caloric values between individuals with a BMI < 25 kg/m^2^ and those classified as obese was 100 kcal, indicating a considerable discrepancy in perceived caloric estimation.

The results of the study also highlighted a general lack of accuracy in caloric estimation among all participants regardless of their BMI. The most accurately estimated meal in terms of caloric content was a “hot dog”, yet only 28.3% of participants correctly identified its caloric value (Figure 2 and Figure 3).

### 3.3. Is the Meal Healthy?

Statistical significance was also observed in participants’ responses to whether they considered a handful of raisins to be a healthy snack. Post hoc tests revealed significant differences between individuals with a BMI < 25 kg/m^2^ and those classified as overweight (*p* = 0.04), as well as between overweight and obese individuals (*p* = 0.009).

A key observation from the study is that most respondents perceived highly processed snacks as unhealthy, while minimally processed snacks were considered healthy. Additionally, the majority of participants classified alcoholic beverages as unhealthy.

### 3.4. A Satiety Duration

A statistically significant difference was observed in the assessment of satiety and the time until hunger was experienced after consuming 500 mL of beer. When asked whether they would feel full after drinking 500 mL of light beer, a significant difference was found between individuals with a BMI < 25 kg/m^2^ and those with a BMI ≥ 30 kg/m^2^ (*p* = 0.034), as well as between individuals with a BMI < 25 kg/m^2^ and those with a BMI of 25–30 kg/m^2^ (*p* = 0.009).

A statistical significance was also observed between individuals with a BMI < 25 kg/m^2^ and those classified as overweight when asked whether they would feel hungry within two hours of consuming 500 mL of beer (*p* < 0.001). Among overweight participants, only one out of 66 reported experiencing hunger within two hours, whereas in the BMI < 25 kg/m^2^ group, 17 out of 92 participants reported feeling hungry within this time frame.

### 3.5. Overestimation of Snack Caloric Content

The tendency to overestimate the caloric content of snacks among individuals with an excessive body mass index (BMI ≥ 25 kg/m^2^) was analyzed. In most cases, individuals with a BMI ≥ 25 kg/m^2^ tended to overestimate the caloric value of snacks; however, none of the tests reached statistical significance. The results are presented in Table 5.

Notably, only chocolate, orange juice, a shot of vodka, and a glass of wine were minimally underestimated by individuals with a BMI ≥ 25 kg/m^2^ (Table 5).

### 3.6. Perception of Portion Sizes and Caloric Value of Snacks and Beverages Depending on Gender

Statistically significant differences were observed in the assessment of portion sizes of highly processed snacks depending on gender (*p* = 0.006). When assessing highly processed snacks, it was found that women assessed them as larger compared to men. Statistical significance was obtained when assessing cookies with cream filling (*p* = 0.002), hot dog (*p* = 0.01), and doughnut (*p* = 0.029). Statistical differences were not observed in the assessment of portion sizes of low-processed snacks and gender (*p* = 0.145). When assessing individual low-processed snacks, statistical significance was not found in most of them, except for the assessment of mini carrots (*p* = 0.002). In all presented cases women assessed the indicated portion as larger compared to men. When assessing the perception of the size of portions of beverages, statistical significance was also observed (*p* < 0.001). In the evaluation of individual drinks, significance was shown for a glass of Coca-Cola (*p* = 0.016) and beer (*p* < 0.001). Females assessed the indicated portions of drinks as “larger” in comparison to men. There was no statistical significance in estimation of caloric value of snacks and beverages depending on gender.

### 3.7. Perception of Portion Sizes and Caloric Value of Snacks and Beverages Depending on Age

No significant differences were demonstrated in the evaluation of the perception of caloric value of highly processed and low-processed snacks and beverages depending on age. Similarly, no statistical significance was demonstrated in the evaluation of portion size depending on this factor.

## 4. Discussion

The results of our study showed no statistical significance in the perception of portion sizes, degree of satiety, or assessment of whether a meal is healthy of most snacks presented in the images. Additionally, no significant difference was found between the estimated caloric values of most of the studied snacks (with three exceptions: milky chocolate, a glass of vodka, and a handful of walnuts). Most respondents considered highly processed meals and alcoholic beverages unhealthy, while vegetables, fruits, and nuts were regarded as healthy snacks. However, no statistical significance was found in the differences in perceptions of healthy/unhealthy snacks between the study groups.

Our study showed that individuals living with overweight or obesity (BMI ≥ 25 kg/m^2^) tend to overestimate caloric values (with exceptions for chocolate, a glass of wine, a shot of vodka, and orange juice), but the results were not statistically significant. This fact may suggest that obese individuals perceive snacks in a very similar way to those of normal weight, or even consider them to be more caloric.

Our results showed that snacks were perceived similarly by obese and non-obese participants in terms of both portion size and satiety. Several factors could influence this result. Some individuals perceive portion sizes through comparisons with the plate size—this is known as the Delboeuf effect, meaning the same meal will be perceived differently depending on whether it is served on a larger or smaller plate [26]. In our study, all meals were served on the same dishware, probably allowing for comparable results across BMI groups. People who perceive portion size also consider their previous experiences of satiety after consuming comparable portions [27]. This aspect was also examined and pointed out in our study. Assessment of satiety after eating the pictured snack was not statistically significant, except for the glass of beer.

In our study patients generally overestimated caloric value. Moreover, individuals with excessive BMI (≥25 kg/m^2^) overestimated meal caloric values even greater than ones with a normal BMI (<25 kg/m^2^). Calorie overestimation may be related to the perception of foods as unhealthy [28]. This may be a reason why respondents who took part in our study tended to overestimate the caloric value of snacks. Analyzing the volunteers’ responses, most snacks were considered as unhealthy meals (Table 2, Table 3 and Table 4), especially high-processed snacks. Another reason for incorrect calorie estimation may be absence of so-called “negative calorie illusion”. It has been proven that presenting unhealthy foods (energy-dense and low in nutrients) next to products considered as healthy foods (e.g., fruit, vegetables) results in lower caloric values, sometimes an underestimation error of up to 100 kcal [29,30]. For example, the estimated caloric value of a fast-food meal presented separately will be higher when presented with vegetables and fruit. In our study, snacks and beverages were always presented separately. This could potentially prevent the error known as the negative calorie illusion and higher caloric values were chosen.

On the other hand, studies have shown that most people tend to underestimate their caloric intake [31]. Based on the authors’ experience, most obese individuals following a reduction diet report consuming approximately 1200 kcal/day. This discrepancy in caloric estimation could explain the difficulty in weight reduction. A similar conclusion was made by Lichtmann et al. In their study, the authors proved that obese people consume more calories than they declare. They also declare more physical activeness than they actually perform [32].

In this study we have proven that individuals with excessive body weight have a greater tendency to overestimate the caloric value of snacks and beverages. The following question therefore arises: Why, despite the presumption that snacks are more caloric, do some people with obesity consume them, knowing that this is leading to an excessive caloric balance? Another question is why they often declare a smaller consumption of calories than exists in reality. There might be many reasons for these phenomena. Perhaps obese patients tend to overestimate snacks, while estimating other (main) meals in a different way—hence the erroneous assessment of daily calorie intake. We reached a similar conclusion in the study assessing the differences in the perception of main meals depending on BMI. In the study participants with excessive body weight tended to underestimate the caloric value of main meals [33]. Another explanation may be that patients with excess body weight, despite overestimating meals, consume them, which affects the positive daily energy balance. It might also be that snacks often have a small “metric volume” and their consumption is “unnoticed” by people with excessive weight. Another reason might be the education status of the respondents taking part in our study. People with higher education might overestimate meals, being more aware of their caloric value and the negative impact on the positive energy balance.

We also pointed out that individuals with excessive weight assess the portion size of snacks similarly to people with normal weight. Similar conclusions were reached by Reily et al., who proved that portion-size preference did not differ depending on BMI [34]. Perhaps there are other factors like meal frequency, food type, or others, which contribute to excessive body weight.

We also found differences in the perception of portion sizes of snacks and drinks depending on gender. The statistical significance described in the results indicates that women perceive the pictured portions as larger compared to men. Higher daily energy intake by men compared to women was also described by Leblanc [35]. Differences in appetite regulation between women and men may result from differences in sex hormones (estrogen reduces appetite, while testosterone increases the amount of food consumed) [36]. An important factor is different metabolic homeostasis in males and females. Moreover, males are more dependent on carbohydrates than females [37]. In addition, it may result from sex differences in basal metabolic rate, which is lower in women than in men [38]. All of the aforementioned issues may cause a different perception of the size of meal portions, probably with a different level of satiety after eating the same portion. No statistically significant differences were demonstrated between the studied subgroups and the assessment of the caloric value of snacks.

Recently, changes in dietary patterns have begun to be under close observation and medical evaluation. Larger portion sizes, an increase in the consumption of snacks, and consequently higher daily energy intake are becoming more significant factors making treatment of obesity difficult [39]. The definition of a snack is not clearly established [40]. A snack is generally considered a meal that is eaten at a specific time of day, depending on the specific type of food, the size of the meal, the place where the food is eaten, or a combination of these factors [6,41,42]. A single snack can provide even 152–302 calories [43,44]. Thus, with multiple snacks consumed daily, total energy intake can increase significantly. It leads to a positive energy balance and contributing to overweight or obesity. Some individuals compensate for the energy surplus from snacks by consuming smaller main meals [45]. According to another study, the energy surplus from snack consumption is not compensated for and contributes to a positive energy balance [46]. A different study by Forslund et al. found that obese patients consume snacks more frequently than individuals of normal weight [47]. On the contrary, the study by Hampl et al. found no relationship between BMI and snacking patterns, including specific snacking times (e.g., evening snacking) [48].

A pivotal aspect in obesity treatment is portion size control. In recent years, there has been a rising trend in the portion sizes of energy-dense foods [49], including both commercially sold food and meals consumed at home. Analyzing cookbooks over the years, it was observed that the average portion size recommended in recipes has increased (Mean calories value of portion suggested in recipes increased by 21% over 100 years) [50]. It has been proven that as portion sizes increase, so does energy intake. A meta-analysis by Zlatevska et al. found that doubling the portion size of a meal leads to a 35% increase in energy intake [51]. It suggests that obese people need more dietetic education with emphasis on proper meal portions. Individual adjustment of the appropriate meal portion size can help to prevent excessive energy intake.

Continuous dietary education and improved caloric estimation skills may help obese patients in their individualized obesity treatment. It has been proven that calorie counting can help with weight loss. Another factor influencing weight loss is professional dietary consultation [52]. It seems reasonable to implement mandatory information on food products and restaurants regarding the calorie content of consumed meals. Hobin et al. conducted a study in which it was found that information about the calorie content of alcoholic and non-alcoholic beverages increased awareness of the calorie content of these beverages. However, there was no significant difference in the change in the amount of drinks ordered or the type of drinks ordered despite their labeling in restaurants [53]. Another study also proved that the adding of labels which inform about calorie content of a meal improves the ability to estimate the calorie value; however, the percentage of people estimating correctly is still small [54]. On the other hand, in our study, no statistically significant differences were observed in the estimation of calories between the study groups. This may suggest that counting calories is not a key element in obesity treatment. The majority of respondents in the study are people with higher education (almost 70% surveyed have academic education). Despite this, most of the participants incorrectly estimated calories, overestimating them. This may suggest that the correct estimation of the caloric value is difficult even for people with higher education. However, in the study conducted by Harnack et al., caloric labeling did not show a significant effect on reducing calorie intake among consumers [55]. Another study provided similar results—that caloric labeling does not help to promote healthier food choices [56]. Better effects of treating individuals living with overweight and obesity may be achieved by introduce information about the recommended daily calorie requirement [57,58]. On the other hand, in the study by Dawns et al., it was found that providing calorie recommendations did not affect the number of calories purchased by the study participants [58]. Researchers studying the influence of the assessment of calorie content on overweight and obesity are still contradictory and more comprehensive studies are required.

In our opinion, dietetic education on the caloric value of meals, daily caloric requirement, and the effects of excessive energy consumption should be conducted already during the school period by schools and pre-school institutions. Dietary habits that were determined in the early years of life often persist in adulthood [19]. This is why emphasis on nutritional education among young people can bring beneficial effects in preventing obesity.

The promotion of healthy snacks, such as vegetables and fruits, is also encouraged [59]. Healthy snacks can be a source of essential micronutrients, vitamins, and fiber [40,59]. Among children, there is a trend of increased consumption of sugar-sweetened beverages, alongside a decrease in the consumption of vegetables and fruits, which can contributes to the development of obesity in this group [17]. This phenomenon may have several causes, including the high availability and palatability of processed snacks, the ease of storage and transport, rising vegetable and fruit prices, and climatic factors limiting the cultivation of fresh produce in certain countries. Fortunately, in recent years, some changes have been observed. Reports from 2024 have shown that young people are becoming more aware of the connection between health and nutrition. Unfortunately, despite their knowledge, the level of healthy snacks consumed (vegetables and fruits) was still insufficient [60]. Another study showed that in some European countries more and more young people chose fruits and vegetables instead of high-processed foods than in the past [18]. Perhaps the higher awareness of unhealthy diet outcomes is one of the reasons.

In summary, the results of our study did not show significant differences in the perception of portions between people with different BMIs. The study participants had a tendency to overestimate caloric value. People with excessive body weight had a greater tendency to overestimate the calories of snacks and drinks. In the treatment of obesity, dietary education is important, with an emphasis on the ability to correctly estimate the calories consumed and to select the appropriate portion size. However, current studies indicate the contemporary results of such education. Dietary education should take place at school age and also concern learning how to choose healthy meals and snacks, which can be a source of fiber and vitamins.

## 5. Conclusions

In general, individuals taking part in our study overestimated the caloric value of snacks. Patients with overweight and obesity tended to overestimate the caloric content of snacks compared to individuals with normal body weight. No difference in the perception of snack portion sizes among patients with different BMIs was noted. Women assessed the portion size of high-processed snacks as larger than men. Additionally, a very low percentage of respondents were able to accurately estimate the caloric content of snacks, regardless of BMI. Generally, patients with different BMIs reported experiencing the same degree of satiety after consuming similar snacks. Therefore, it seems that obese patients experience a similar level of fullness after eating snacks and beverages and estimate the caloric content of snacks and drinks similarly to those with normal weight.

## 6. Limitations

According to the authors, an important limitation of the study is the inability to verify the data regarding the weight and height of the participants. Volunteers who took part in the study completed the survey online, making it impossible to accurately assess these parameters. Another limitation is the relatively small number of participants in the study.

Another limitation of the study is that the survey was available not only for patients in hospitals and outpatients clinic but also on websites and social media dedicated to nutrition and healthy lifestyle. In our study, we considered the influence of BMI on the respondents’ answers. It should be remembered that other variables that were not included in the study as exclusion criteria may also have an impact on the volunteers’ answers. Such parameters include, for example, the time since the last meal before filling the questionnaire, the presence of comorbidities, medications taken, and daily physical activity.

## 7. Strengths

The sample size of the study allowed for reliable results. The study addresses the impact of BMI on the perception of portion sizes and the degree of satiety. This publication is one of the few that explores such a topic. As one of the rare studies to do so, it describes the relationship between BMI and the perception of meal portion sizes.

## Figures and Tables

**Figure 1 nutrients-17-02123-f001:**
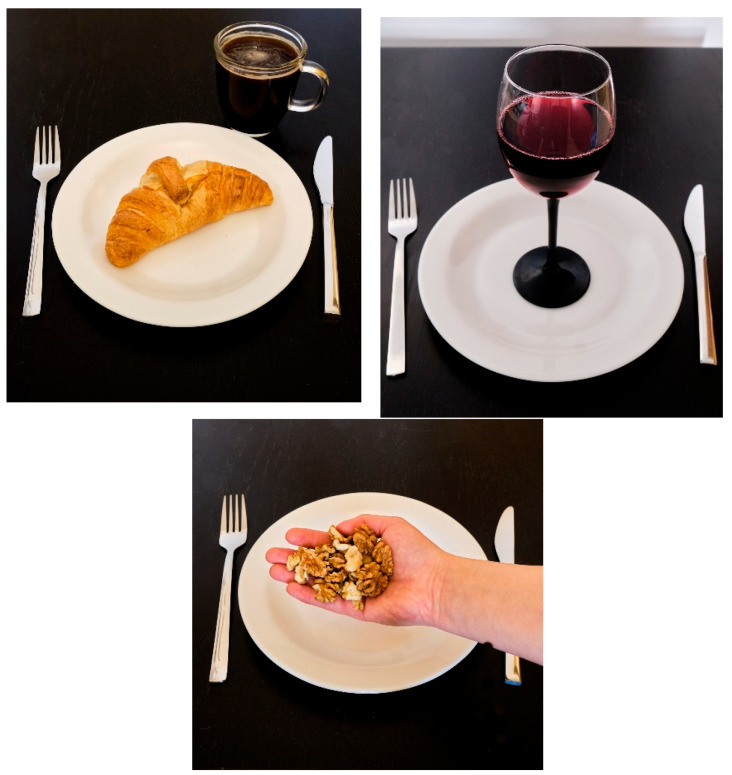
Examples of pictures used in questionnaire (croissant with black coffee without sugar, red wine, handful of walnuts).

**Figure 2 nutrients-17-02123-f002:**
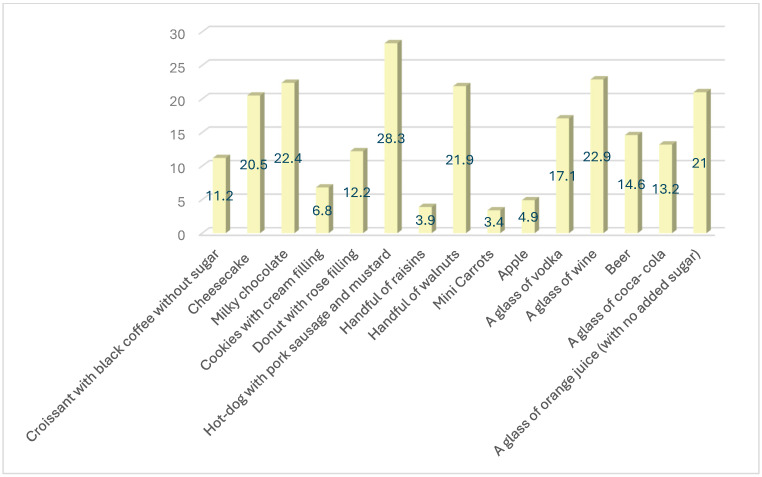
Percentage of participants who accurately (90–110% exact caloric value) estimated caloric value of meals.

**Figure 3 nutrients-17-02123-f003:**
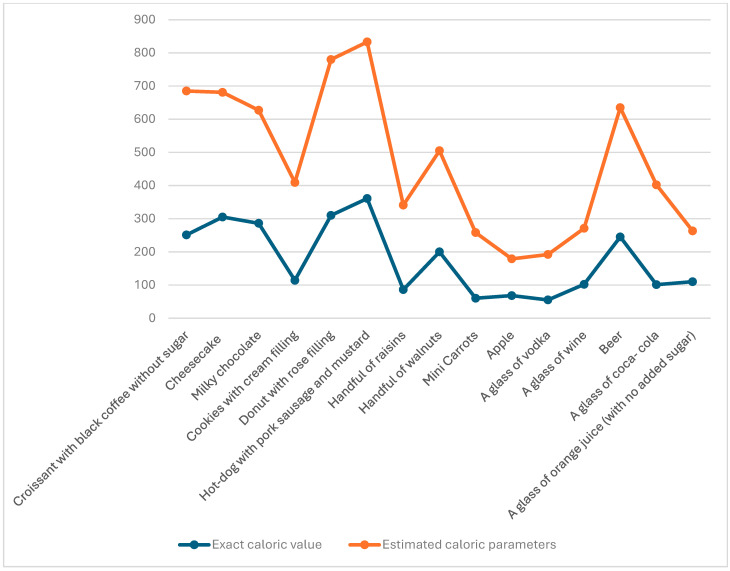
Exact calorie content compared to participant average rating.

**Table 1 nutrients-17-02123-t001:** Details of the study group.

Variable		N [%] or M ± SD
Gender	Female	149 [72.7]
	Male	56 [27.3]
Age	(years)	39.7 ± 13.1
Weight	(kg)	78.2 ± 20.9
Height	(cm)	169.9 ± 8.6
BMI	(kg/m^2^)	27.0 ±6.3
BMI (ranges) [%]	<25	92 [44.9]
	25–29.9	66 [30.7]
	≥30	47 [24.4]
Education N [%]	Academic	142 [69.3]
	High school	62 [30.2]
	Primary education	1 [0.5]

**Table 2 nutrients-17-02123-t002:** Highly processed snacks.

Feature	Description	BMI			*p*
		<25 (n = 92)	25–29.9 (n = 66)	≥30 (n = 47)	
Croissant with black coffee without sugar (251 kcal)					
Meal size [1–10]	Med. (IQR)	4 (2)	3 (2)	4 (2)	0.691
Fullness feeling [1–10]	Med. (IQR)	4 (2)	4 (2)	4 (3)	0.977
Estimated calorie value [kcal]	Med. (IQR)	350 (250)	300 (200)	375 (200)	0.178
“Do you think this meal is healthy?”	Y/N	14/78	8/58	10/37	0.413
“Will you feel full after eating this meal?”	Y/N	52/40	48/18	28/19	0.150
“Will you NOT feel hunger/craving in the next 2 h?”	Y/N	21/71	8/58	8/39	0.221
Cheesecake (305 kcal)					
Meal size [1–10]	Med. (IQR)	4 (2)	4 (2)	4 (2)	0.837
Fullness feeling [1–10]	Med. (IQR)	4 (2)	4 (2)	4 (2)	0.917
Estimated calorie value [kcal]	Med. (IQR)	350 (200)	300 (150)	320 (200)	0.637
“Do you think this meal is healthy?”	Y/N	9/83	9/57	4/43	0.635
“Will you feel full after eating this meal?”	Y/N	63/29	47/19	33/14	0.931
“Will you NOT feel hunger/craving in the next 2 h?”	Y/N	12/80	9/57	5/42	0.886
Milky chocolate (286 kcal)					
Meal size [1–10]	Med. (IQR)	3 (3)	4 (2)	4 (2)	0.376
Fullness feeling [1–10]	Med. (IQR)	3 (2)	3 (3)	3 (3)	0.887
Estimated calorie value [kcal]	Med. (IQR)	300 (200)	250 (200)	300 (250)	**0.026**
“Do you think this meal is healthy?”	YN	9/83	5/61	4/43	0.887
“Will you feel full after eating this meal?”	Y/N	67/25	49/17	30/17	0.436
“Will you NOT feel hunger/craving in the next 2 h?”	Y/N	8/84	1/65	5/42	0.105
Cookies with cream filling (114 kcal)					
Meal size [1–10]	Med. (IQR)	3 (2)	3 (2)	3 (3)	0.345
Fullness feeling [1–10]	Med. (IQR)	3 (2)	3 (2)	3 (2)	0.582
Estimated calorie value [kcal]	Med. (IQR)	245 (170)	250 (120)	300 (200)	0.124
“Do you think this meal is healthy?”	YN	1/91	1/65	0/47	0.714
“Will you feel full after eating this meal?”	Y/N	70/22	53/13	36/11	0.809
“Will you NOT feel hunger/craving in the next 2 h?”	Y/N	61/31	46/20	30/17	0.800
Donut with rose filling (310 kcal)					
Meal size [1–10]	Med. (IQR)	5 (4)	5 (2)	4 (4)	0.849
Fullness feeling [1–10]	Med. (IQR)	5 (4)	5 (4)	4 (4)	0.373
Estimated calorie value [kcal]	Med. (IQR)	400 (200)	400 (150)	400 (150)	0.975
“Do you think this meal is healthy?”	YN	1/92	0/66	0/47	0.539
“Will you feel full after eating this meal?”	Y/N	32/60	27/39	24/23	0.180
“Will you NOT feel hunger/craving in the next 2 h?”	Y/N	22/70	8/58	11/36	0.151
Hot dog with pork sausage and mustard (361 kcal)					
Meal size [1–10]	Med. (IQR)	6 (3)	6 (2)	5 (4)	0.772
Fullness feeling [1–10]	Med. (IQR)	6 (4)	6 (2)	6 (3)	0.996
Estimated calorie value [kcal]	Med. (IQR)	400 (213)	400 (200)	400 (150)	0.970
“Do you think this meal is healthy?”	YN	3/89	5/61	3/44	0.465
“Will you feel full after eating this meal?”	Y/N	23/69	11/55	10/37	0.453
“Will you NOT feel hunger/craving in the next 2 h?”	Y/N	52/40	28/38	22/25	0.195

Bold indicates *p*-value < 0.05.

**Table 3 nutrients-17-02123-t003:** Low-processed snacks.

**Feature**	**Description**	**BMI**			** *p* **
		**<25 (n = 92)**	**25–29.9 (n = 66)**	**≥30 (n = 47)**	
Handful of raisins (86 kcal)					
Meal size [1–10]	Med. (IQR)	3 (3)	3 (3)	4 (3)	0.074
Fullness feeling [1–10]	Med. (IQR)	3 (2)	3 (3)	3 (3)	0.513
Estimated calorie value [kcal]	Med. (IQR)	200 (200)	200 (170)	250 (200)	0.219
“Do you think this meal is healthy?”	Y/N	57/35	30/36	33/14	**0.021**
“Will you feel full after eating this meal?”	T/N	70/22	50/16	36/11	0.995
“Will you NOT feel hunger/craving in the next 2 h?”	T/N	7/85	6/60	4/70	0.382
Handful of walnuts (200 kcal)					
Meal size [1–10]	Med. (IQR)	4 (2)	4 (2)	5 (3)	0.264
Fulness feeling [1–10]	Med. (IQR)	4 (4)	4 (2)	4 (3)	0.364
Estimated calorie value [kcal]	Med. (IQR)	250 (250)	210 (220)	350 (250)	**0.031**
“Do you think this meal is healthy?”	Y/N	90/2	65/1	47/0	0.600
“Will you feel full after eating this meal?”	Y/N	54/47	29/37	25/22	0.617
“Will you NOT feel hunger/craving in the next 2 h?”	Y/N	14/78	13/53	10/37	0.622
Mini carrots (60 kcal)					
Meal size [1–10]	Med. (IQR)	4 (3)	4 (3)	5 (3)	0.440
Fullness feeling [1–10]	Med. (IQR)	4 (2)	4 (3)	4 (3)	0.860
Estimated calorie value [kcal]	Med. (IQR)	150 (140)	150 (130)	150 (170)	0.064
“Do you think this meal is healthy?”	Y/N	90/2	63/3	46/1	0.638
“Will you feel full after eating this meal?”	Y/N	46/46	33/33	20/27	0.669
“Will you NOT feel hunger/craving in the next 2 h?”	Y/N	16/76	8/58	9/38	0.546
Apple (68 kcal)					
Meal size [1–10]	Med. (IQR)	3 (2)	3 (2)	4 (2)	0.135
Fullness feeling [1–10]	Med. (IQR)	3 (2)	3 (1)	4 (3)	0.334
Estimated calorie value [kcal]	Med. (IQR)	80 (70)	100 (100)	80 (100)	0.378
“Do you think this meal is healthy?”	Y/N	90/2	64/2	47/0	0.506
“Will you feel full after eating this meal?”	Y/N	60/32	42/24	26/21	0.507
“Will you NOT feel hunger/craving in the next 2 h?”	Y/N	4/88	3/63	3/44	0.860

Y—yes, N—no, Med.—median, IQR—interquartile range. Bold indicates *p*-value < 0.05.

**Table 4 nutrients-17-02123-t004:** Drinks.

Feature	Description	BMI			*p*
		<25 (n = 92)	25–29.9 (n = 66)	≥30 (n = 47)	
A glass of vodka (55 kcal)					
Meal size [1–10]	Med. (IQR)	1 (1)	1 (1)	1 (2)	0.728
Fullness feeling [1–10]	Med. (IQR)	1 (0)	1 (0)	1 (0)	0.878
Estimated calorie value [kcal]	Med. (IQR)	100 (68)	100 (138)	100 (180)	**0.011**
“Do you think this meal is healthy?”	Y/N	4/88	0/66	2/45	0.230
“Will you feel full after eating this meal?”	Y/N	86/6	63/3	45/2	0.801
“Will you NOT feel hunger/craving in the next 2 h?”	Y/N	3/89	2/64	1/46	0.930
A glass of wine (102 kcal)					
Meal size [1–10]	Med. (IQR)	3 (3)	2 (2)	3 (3)	0.746
Fullness feeling [1–10]	Med. (IQR)	2 (2)	1 (2)	1 (1)	0.454
Estimated calorie value [kcal]	Med. (IQR)	120 (100)	100 (111)	150 (200)	0.098
“Do you think this meal is healthy?”	Y/N	24/68	24/42	18/29	0.235
“Will you feel full after eating this meal?”	Y/N	83/9	61/5	41/6	0.657
“Will you NOT feel hunger/craving in the next 2 h?”	Y/N	2/90	2/64	2/45	0.787
Beer (245 kcal)					
Meal size [1–10]	Med. (IQR)	5 (5)	5 (4)	5 (5)	0.953
Fullness feeling [1–10]	Med. (IQR)	4 (4)	3 (3)	3 (3)	0.258
Estimated calorie value [kcal]	Med. (IQR)	300 (200)	300 (200)	300 (250)	0.346
“Do you think this meal is healthy?”	YN	2/90	4/62	0/47	0.144
“Will you feel full after eating this meal?”	Y/N	56/36	53/13	37/10	**0.013**
“Will you NOT feel hunger/craving in the next 2 h?”	Y/N	17/75	1/65	3/44	**0.001**
A glass of Coca-Cola (101 kcal)					
Meal size [1–10]	Med. (IQR)	3 (3)	3 (2)	3 (3)	0.343
Fullness feeling [1–10]	Med. (IQR)	2 (3)	2 (2)	2 (2)	0.982
Estimated calorie value [kcal]	Med. (IQR)	200 (200)	200 (150)	200 (180)	0.757
“Do you think this meal is healthy?”	YN	2/90	2/64	5/42	0.057
“Will you feel full after eating this meal?”	Y/N	81/11	59/7	42/5	0.955
“Will you NOT feel hunger/craving in the next 2 h?”	Y/N	2/92	2/64	4/43	0.171
A glass of orange juice (with no added sugar) (110 kcal)					
Meal size [1–10]	Med. (IQR)	3 (3)	3 (2)	3 (3)	0.296
Fullness feeling [1–10]	Med. (IQR)	2 (2)	2 (3)	3 (3)	0.328
Estimated calorie value [kcal]	Med. (IQR)	120 (70)	100 (70)	150 (120)	0.272
“Do you think this meal is healthy?”	YN	74/18	56/10	41/6	0.553
“Will you feel full after eating this meal?”	Y/N	73/19	54/12	36/11	0.794
“Will you NOT feel hunger/craving in the next 2 h?”	Y/N	2/90	1/65	4/43	0.088

Bold indicates *p*-value < 0.05.

**Table 5 nutrients-17-02123-t005:** Odds ratio for overestimation of the meals’ calorie value in individuals with BMI ≥ 25 kg/m^2^.

Meal	OR	CI (0.95)	*p*
Croissant with black coffee without sugar	1.14	0.61–2.12	0.789
Cheesecake	1.06	0.61–1.84	0.889
Milky chocolate	0.90	0.52–1.57	0.778
Cookies with cream filling	1.26	0.54–2.96	0.665
Donut with rose filling	1.32	0.7–2.48	0.422
Hot dog with pork sausage and mustard	1.09	0.62–1.91	0.775
Handful of raisins	1.26	0.52–3.05	0.625
Handful of walnuts	1.4	0.81–2.44	0.26
Mini carrots	1.55	0.8–3.0	0.236
Apple	1.21	0.7–2.12	0.570
A glass of vodka	0.71	039–1.29	0.291
A glass of wine	0.97	0.56–1.69	1.0
Beer	1.28	0.7–2.33	0.445
A glass of Coca-Cola	1.54	0.81–2.94	0.193
A glass of orange juice (with no added sugar)	0.98	0.57–1.7	1.0

OR—odds ratio, CI—confidence interval, *p*—*p*-value.

## Data Availability

The data presented in this study are available on request from the corresponding author for privacy reasons.

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
