# Peer review of "Differences in the Perception of Snacks and Beverages Portion Sizes Depending on Body Mass Index"

_nutrients, 2025, doi:10.3390/nu17132123_

Round 1

Reviewer 1 Report

Comments and Suggestions for Authors

This manuscript explores whether BMI affects the perception of portion size, estimation of caloric content and estimated satiety after consumption of a range of foods and drinks. The study aims to address the potential link between snack consumption and BMI, however there are some major improvements needed, suggestions are listed below. Overall, particularly for the introduction, there is a lack of referencing of statements. There are grammatical errors and typos throughout. The discussion includes some links to existing literature, however there is little in terms of the future impact of the work and the strengths and limitations are too brief. Ethics statement and data availability statement are not given.

Abstract – there should be a clear statement of the study aims/objectives.

Introduction:

Line 37-38: Do you have some evidence to indicate that snacks tend to be high in saturated fats and carbohydrates? Is it the same in all countries?

Line 38-39: I think the sentence regarding snacks being associated with a sense of reward, needs a reference to back up.

Line 40-41: Again, I think there needs to be a reference to support this statement, that snacks contribute a significant proportion of total energy intake. E.g. is this national consumption data? If so, which country?

Line 47-49: This seems an oversimplification of the issue and I’m not sure if accurate. Do you have a reference to support this?

Materials and methods:

Lines 69-71: The questionnaire was available on healthy nutrition and weight management  websites – biased population? Is this mentioned in limitations? Why was it decided to promote  here?

Lines 79-80: This sentence (first sentence in paragraph) should be in results section.

Line 85: Please can provide more detail on how the snacks were chosen?

Line 102: Who categorised these as snacks?

Line 129-130: This sentence doesn’t make sense, I think it needs explaining further.

Line 134: Table 1 should be in the results section. The results section should begin with a  description of the study population.

Line 139:  spelling error- red ‘vine’ should be changed to ‘wine’

Line  194: Table 2 is very large and there are few significant findings. I wonder if it can be reduced to just the significant findings and the remaining added as a table in supplementary material?

Line 202: The term ‘patients’ should be replaced with ‘participants’. Also, somewhere in methods – in statistical analysis section perhaps – you need to add how you calculated the percentage of participants who accurately estimated the caloric value of foods and drinks – emphasising  that 90-110% was considered accurate.

Figure 2: Repetition of title in figure and below figure. Recommend removing from within figure.

Figure 3: There is a typo in the title (“comparted”) and overall the title does not make sense. Should it read “Exact caloric content compared to average participant estimation”? I’m also not convinced you need both figure 3 and table 6 as both showing the same findings. 

Line 189 – I recommend removing the term ‘abnormal’

Discussion:

Line 210: Change term to ‘individuals living with overweight and obesity’

Lines 220-225: These sentences do not read well; I recommend a re-write to make the flow easier for the reader.

Lines 252-259: These sentences need reviewing for grammar, they don’t make sense in parts, e.g. ‘….in their case individually to gain proper energy balance.’

Line 283: ‘Continuous dietary education and improved caloric estimation skills may help obese patients…’ – How do you see envisage this happening? Can you elaborate? It seems that the study population were already well educated. In the UK, calorie information is now present on all menus in restaurants, fast food takeaways etc, are there any studies that show the effectiveness of this policy in improving calorie estimation?

Strengths and limitations: this section is far too brief, particularly for limitations. Some to consider:

  • How generalisable are the study findings to the wider population? Has the recruitment method limited this?
  • Perhaps the length of time since last meal would affect the answers given in terms of satiety?
  • There are papers on the portion estimation – real vs on-screen images – and the effect on accuracy. I think this literature needs referencing. In addition, are there any studies looking at perceived satiety from images rather than real-life setting?
Comments on the Quality of English Language

There are grammatical errors and typos throughout, which need addressing. 

Author Response

Dear Reviewer 1,

First of all, we would kindly like to thank you for the review. We have corrected the manuscript according to all your valuable comments and suggestions. We hope the corrected manuscript will meet your all expectations. Below, we have attached the answers for all of your questions and suggestions.

All changes and updates were marked red in main the text.

This manuscript explores whether BMI affects the perception of portion size, estimation of caloric content and estimated satiety after consumption of a range of foods and drinks. The study aims to address the potential link between snack consumption and BMI, however there are some major improvements needed, suggestions are listed below.

  1. Overall, particularly for the introduction, there is a lack of referencing of statements.
    Text updated, references added to section Introduction.
  2. There are grammatical errors and typos throughout.
    Text updated and visiable errors corrected.
  3. The discussion includes some links to existing literature, however there is little in terms of the future impact of the work and the strengths and limitations are too brief.
    Text updated.
  4. Ethics statement and data availability statement are not given.
    Text updated – paragraph added.
  5. Abstract – there should be a clear statement of the study aims/objectives.

Abstract updated. 

  1. Introduction: Line 37-38: Do you have some evidence to indicate that snacks tend to be high in saturated fats and carbohydrates? Is it the same in all countries?
    Text updated, references added.
  1. Line 38-39: I think the sentence regarding snacks being associated with a sense of reward, needs a reference to back up.
    Text updated, references added.
  1. Line 40-41: Again, I think there needs to be a reference to support this statement, that snacks contribute a significant proportion of total energy intake. E.g. is this national consumption data? If so, which country?
    Text updated, references added.
  2. Line 47-49: This seems an oversimplification of the issue and I’m not sure if accurate. Do you have a reference to support this?
    Text updated, references added.
  3. Materials and methods: Lines 69-71: The questionnaire was available on healthy nutrition and weight management  websites – biased population? Is this mentioned in limitations? Why was it decided to promote  here?
    Limitations have been updated. We inititally tried to recruit patients in hospital and outpatient clinics, however small interest of patients lead us to this „open-type” of recruitment. We decided to use it, as we believed it would help with recruitment of large population better reflecting the population in our country. By making the questionaire available on websites and social media dedicated to nutrition and weight management, we were able to obtain the final numer of paricipants (>200).
  4. Lines 79-80: This sentence (first sentence in paragraph) should be in results section.
    Text updeted.
  5. Line 85: Please can provide more detail on how the snacks were chosen?
    Text updated. More details are added.
  1. Line 102: Who categorised these as snacks?
    Text updated. Definition of snack is ambigious. We added criteria which we have chosen and gave some references which support our definition.
  1. Line 129-130: This sentence doesn’t make sense, I think it needs explaining further.

Dear reviewer, could you kindly specify is there grammatical or methological error in the text? We would improve it with your valuable comment. Lines which you indicated are in section Statistical Analysis.

  1. Line 134: Table 1 should be in the results section. The results section should begin with a  description of the study population.
    Text updated
  1. Line 139:  spelling error- red ‘vine’ should be changed to ‘wine’
    Text updated
  2. Line  194: Table 2 is very large and there are few significant findings. I wonder if it can be reduced to just the significant findings and the remaining added as a table in supplementary material?
    We wanted to include all the information to present the results in a more detailed and comprehensive way. Significant findings are described in the text
  3. Line 202: The term ‘patients’ should be replaced with ‘participants’. Also, somewhere in methods – in statistical analysis section perhaps – you need to add how you calculated the percentage of participants who accurately estimated the caloric value of foods and drinks – emphasising  that 90-110% was considered accurate.
    Text updated
  1. Figure 2: Repetition of title in figure and below figure. Recommend removing from within figure.
    Figure updated
  2. Figure 3: There is a typo in the title (“comparted”) and overall the title does not make sense. Should it read “Exact caloric content compared to average participant estimation”? I’m also not convinced you need both figure 3 and table 5 as both showing the same findings. 
    We have corrected the title of Figure 3. Figure 3 shows the general tendency for all study participants to misjudge the caloric value of snacks. While table 5 shows the tendency for individuals with excessive BMI to overestimate the caloric content of snacks compared to normal weight individuals. Both pieces of information are valuable because one shows how poorly people are able to correctly estimate snacks in general, and the other gives emphasis on how overweight and obese individuals perceive the caloric content of snacks.
  3. Line 189 – I recommend removing the term ‘abnormal’
    Text updated
  1. Discussion: Line 210: Change term to ‘individuals living with overweight and obesity’
    Text updated
  2. Lines 220-225: These sentences do not read well; I recommend a re-write to make the flow easier for the reader.
    Text updated
  3. Lines 252-259: These sentences need reviewing for grammar, they don’t make sense in parts, e.g. ‘….in their case individually to gain proper energy balance.’
    Text updated
  4. Line 283: ‘Continuous dietary education and improved caloric estimation skills may help obese patients…’ – How do you see envisage this happening? Can you elaborate? It seems that the study population were already well educated. In the UK, calorie information is now present on all menus in restaurants, fast food takeaways etc, are there any studies that show the effectiveness of this policy in improving calorie estimation?
    Paragraph in section Discusion is added.
  5. Strengths and limitations: this section is far too brief, particularly for limitations. Some to consider: How generalisable are the study findings to the wider population? Has the recruitment method limited this?
    Text updated.
  6. Perhaps the length of time since last meal would affect the answers given in terms of satiety?
    Text updated, more limitations added
  7. There are papers on the portion estimation – real vs on-screen images – and the effect on accuracy. I think this literature needs referencing. In addition, are there any studies looking at perceived satiety from images rather than real-life setting?
    We have evaluated the available databases in order to prepare comprehensive discussion. If there is a sugestion of additional study to be discuted we are open for suggestions, in order to expand discussion and our article value.

Reviewer 2 Report

Comments and Suggestions for Authors

Dear authors,

I have read your paper entitled “Differences in the perception of snack and beverages portion 2 sizes depending on body mass index” written by Anna Celina Durmaet al.

The title is explicit and relevant for the study.

The abstract follows the structure established by the Nutrients journal.

The introduction provides sufficient information regarding the actual knowledge of obesity and excessive energy intake due to different types of snack consumption.

The Materials and Methods section is well described, including the number of participants included in the study and how the questionnaire was conducted. Moreover, it includes the sections of the questionnaire and their detailed description.

The “Results” Section is missing and should be added (title).

The study contains numerous data that compared the different types of snacks, the meal size and the BMI.

The discussion section is comprehensive, but I believe more studies should be cited in order to increase the quality of the article. Maybe the number of steps that the participant is performing daily? Or any type of exercises? Does the physical activity influence the overestimation of the food intake? Moreover, are your data different between the age groups, gender, age group and gender?

Regarding the discussion section, the last paragraph should be written clearer and more concise. The essential information is hard to retain.

I have read your questionnaire. It is very interesting and easy to complete. More correlations should be made between all the parameters evaluated in order to increase the quality of the article.

This research can further open new perspectives for future studies, due to the fact that more and more people are overweight or obese. Informing population regarding healthy snack eating should be promoted on all types of media.

The references are relevant and aligned with the topic of the study.

Moreover, a graphical abstract would be appreciated.

Author Response

Dear Reviewer 2,

First of all, we would kindly like to thank you for the review. We have corrected the manuscript according to all your valuable comments and suggestions. We hope the corrected manuscript will meet all your expectations. Below, we have attached the answers for all of your questions and suggestions.

All changes and updates were marked green in the main text.

Text updated.  Discussion section was rewritten. The correlations you mention are a very interesting idea. In our manuscript, we focused on the influence of BMI on the perception of meal portions and their caloric value. However, if you think it is justified, we will improve our article with the exact correlation you will indicate. If they are recommended or necessary please indicate them directly , however we will need a little more time than given 7 days to prepare calculations and clear data presentation.

Round 2

Reviewer 2 Report

Comments and Suggestions for Authors

Dear authors,

If it is possible, any further data that can be provided, even in 7 days, could improve the manuscript. Either, I believe that the manuscript has improved significantly.

Moreover, I also believe that a graphical abstract should be provided. This would increase the number of reads and citations to your paper.

Author Response

Dear authors,

If it is possible, any further data that can be provided, even in 7 days, could improve the manuscript. Either, I believe that the manuscript has improved significantly.

Moreover, I also believe that a graphical abstract should be provided. This would increase the number of reads and citations to your paper.

Dear Reviewer 2, We would like again to thank you for your comments and time spent on the manuscript evaluation. Thank you again for giving us the oportunity to revise and improve our article. All your suggestions were considered and included in the manuscript text with red colour.

We added graphical abstract and new paragraph in text. We calculated and described results of comparison portion assesment anc caloric value assesment in relations to gender and age.